# Translational Applications of Linear and Circular Long Noncoding RNAs in Endometriosis

**DOI:** 10.3390/ijms221910626

**Published:** 2021-09-30

**Authors:** Xiyin Wang, Luca Parodi, Shannon M. Hawkins

**Affiliations:** 1Department of Obstetrics and Gynecology, Indiana University School of Medicine, Indianapolis, IN 46202, USA; wang.xiyin@mayo.edu; 2Mayo Clinic Graduate School of Biomedical Sciences, Rochester, MN 55905, USA; 3Obstetrics and Gynecology Department, Istituto Clinico Sant’Anna, 25127 Brescia, Italy; luca.parodi@grupposandonato.it

**Keywords:** endometriosis, human disease, noncoding RNA (ncRNA), long noncoding RNA (lncRNA), circular lncRNA (circRNA)

## Abstract

Endometriosis is a chronic gynecologic disease that negatively affects the quality of life of many women. Unfortunately, endometriosis does not have a cure. The current medical treatments involve hormonal manipulation with unwanted side effects and high recurrence rates after stopping the medication. Sadly, a definitive diagnosis for endometriosis requires invasive surgical procedures, with the risk of complications, additional surgeries in the future, and a high rate of recurrence. Both improved therapies and noninvasive diagnostic tests are needed. The unique molecular features of endometriosis have been studied at the coding gene level. While the molecular components of endometriosis at the small RNA level have been studied extensively, other noncoding RNAs, such as long intergenic noncoding RNAs and the more recently discovered subset of long noncoding RNAs called circular RNAs, have been studied more limitedly. This review describes the molecular formation of long noncoding and the unique circumstances of the formation of circular long noncoding RNAs, their expression and function in endometriosis, and promising preclinical studies. Continued translational research on long noncoding RNAs, including the more stable circular long noncoding RNAs, may lead to improved therapeutic and diagnostic opportunities.

## 1. Introduction

Endometriosis is a progressive and debilitating gynecologic disease whereby endometrial-like tissue grows outside the uterine cavity, invading adjacent organs, such as the ovaries, bladder, colon, or pelvic peritoneum [1,2,3]. Endometriosis is often accompanied by chronic pelvic pain, dysmenorrhea, dyspareunia, dysuria, and dyschezia and can cause infertility [4,5]. The prevalence of endometriosis ranges from 5 to 15% of reproductive age women depending on the method of disease confirmation [3,6], affecting approximately 190 million women worldwide and 5 million women and adolescent girls in the United States [2].

Diagnosing endometriosis is exceptionally challenging since it shares non-specific symptoms, such as pelvic pain, with other conditions. The first-line imaging modality is typically pelvic ultrasound as it can allow for the diagnosis of other conditions that cause pelvic pain. However, the sensitivity and specificity of ultrasound are dependent on endometriosis lesion type and location. For example, transvaginal ultrasonography has high sensitivity and high specificity for ovarian endometriomas [7]. However, the diagnosis of deep infiltrating endometriosis lesions by ultrasound is related to operator expertise [7,8]. Surgical visual inspection by laparoscopy with histologic confirmation is currently the only way to diagnose pathology-proven endometriosis [5]. Surgery involves the risk of surgical complications, adhesion formation, and the need for future surgeries [9]. Because there is no gold standard for noninvasive diagnosis, there is often a significant delay in diagnosis [5]. The median time is seven years from the onset of symptoms with pain or infertility to endometriosis diagnosis [10]. Better diagnostic strategies must be developed.

Unfortunately, there is no cure for endometriosis. Non-steroidal anti-inflammatory medications are routinely used, but they are not more effective than a placebo [11]. Hormonal therapies, including gonadotropin-releasing hormone agonists and newer antagonists, can be prescribed only for a short time because of undesirable side effects, including irregular menstrual bleeding, the development of menopausal symptoms, and the detrimental impact on bone density [7,9,12]. Moreover, both medical and surgical therapies fail to prevent recurrence [2] as 20–50% of endometriosis recurs within five years of treatment [13]. The economic burden of endometriosis in the United States is estimated at $78 billion per year, including direct healthcare costs and indirect costs to patients [14]. Better treatment options are warranted.

Noncoding RNAs (ncRNAs), including microRNAs (miRNAs), long noncoding RNAs (lncRNAs), and circular RNAs (circRNAs), encompass large segments of the transcriptome that do not have apparent protein-coding roles [15]. ncRNAs are divided into two subclasses based on size: short ncRNAs and long ncRNAs. LncRNAs are commonly defined as transcripts longer than 200 nucleotides. Short ncRNAs, including the ~22 nucleotide long miRNAs, have emerged as critical post-transcriptional regulators of gene expression that are fundamental for many disease processes [16]. Although numerous studies have investigated the potential roles of miRNAs as diagnostic biomarkers, no particular miRNA has been translated from bench to clinic for diagnostic purposes [17]. The role of miRNAs in endometriosis has been recently reviewed [18] and will not be included again here.

Similar to messenger RNAs, most lncRNAs are transcribed by RNA polymerase II, and then they undergo post-transcription processes, leading to a 5′ cap, alternative splicing, and 3′ poly(A) tail [19]. Some lncRNAs have been re-defined as protein-encoding genes by closer inspection of the transcriptome and proteome with next-generation sequencing and mass spectrometry [20]. lncRNAs represent the largest class of ncRNAs as over 60,000 lncRNAs have been identified [21]. Many lncRNAs have substantial roles in several biological processes, including endometriosis [22]. As a unique subset of lncRNAs, circRNAs contain a circular secondary structure, characterized by a covalently closed continuous loop structure without 5′-3′ polarity or a poly(A) tail [23]. CircRNAs are involved in the pathogenesis of many diseases [24]. This review will discuss the fundamental roles of both linear and circular long noncoding RNAs in the molecular features of endometriosis and their relevance to current clinical practice. We will also discuss how these preclinical insights into ncRNA biology could develop into diagnostics and therapies in endometriosis.

## 2. Biogenesis, Structure, and Function of Linear and Circular lncRNAs

While more than 90% of the genome is transcribed into RNA, only about 2–5% of that genome contains protein-coding potential [25,26]. The remaining transcriptome comprises ncRNAs transcripts, consisting of small ncRNAs and lncRNAs and both linear and circular lncRNAs. To date, the GENCODE project has conservatively annotated the human genome and believes that it contains 19,954 protein-coding genes and 40,293 noncoding RNA genes [27]. Some 45% (17,957) of the noncoding RNA species are considered lncRNAs that give rise to more than 48,000 distinct transcripts [27]. CircRNAs are different from other long noncoding RNAs due to their single-stranded, circular secondary structure derived from the back splicing of exons from mRNAs and antisense RNAs [28]. In total, there are more than two million circRNAs present in all of the databases [29].

### 2.1. Biogenesis, Structure, and Function of Linear lncRNAs

The biogenesis of lncRNAs is like mRNA biogenesis since this process is mediated by RNA polymerase II. Similar to mRNA molecules, lncRNAs are characterized by alternative splicing, a 5′ 7-methylguanosine cap, and a 3′ poly (A) tail, although there is evidence that certain lncRNAs lack the 5′-cap or 3′poly (A) tail [30]. A comparison of the global features of lncRNAs and mRNAs shows that lncRNAs are less abundantly expressed, have less stability, are less evolutionarily conserved, and contain fewer numbers but longer exons [31]. The expression of lncRNAs is considered more cell- or tissue-specific than mRNA expression [32].

lncRNAs can be transcribed from the intergenic, exonic, or distal protein-coding regions of the genome. Based on genomic locations and orientation, lncRNAs can be classified into intergenic, intronic, sense, and antisense lncRNAs [33]. Intergenic lncRNAs (long intergenic noncoding RNAs or lincRNAs) are located between two protein-coding genes and transcribed in the same direction as those genes. Intronic lncRNAs are located entirely within the intronic region of a protein-coding gene and do not overlap with any exon. Sense lncRNAs are transcribed from the same strand and in the same direction as the protein-coding gene, possibly being exonic and/or intronic. Antisense lncRNAs are transcribed from the opposite strand of the protein-coding gene and can also be exonic and/or intronic. Pseudogene-derived RNAs are key components of lncRNAs with important functions in multiple biological processes [34].

Functionally, lncRNAs are classified under the mechanisms of action into these categories: signals (gene activators), decoys or sponges (gene repressors), guides (gene expression regulators), scaffolds (chromatin modifiers), and enhancer RNAs (eRNAs) [31,35]. As signals, lncRNAs function alone or combined with transcription factors or signaling pathways to activate transcriptional activity in time and space. As decoys, lncRNAs bind to functional sites to titrate the transcription factors away from chromatin or titrate miRNAs away from their targets to modulate transcription. As guides, lncRNAs recruit regulatory proteins to form ribonucleoprotein (RNP) complexes and direct them to their target sites to regulate the expression of target genes, either in cis or in trans. As scaffolds, lncRNAs provide platforms to bring different proteins together to form RNP complexes to activate or repress transcription. eRNAs are regulatory sequences from enhancer regions that in cis regulate the expression of target genes. LncRNAs can be nuclear, cytoplasmic, or both, and the subcellular localization determines its function [31,35].

### 2.2. CircRNA Biogenesis and Structure

CircRNAs are a unique subtype of lncRNAs that are covalently closed, single-stranded circular transcripts without 5′ caps or 3′ poly(A) tails. Although circRNAs were first described in the late 1970s, research in the past decade has dramatically improved our understanding of the expression and various biological functions due to the application of new technologies, mainly deep RNA sequencing [36]. The single-stranded, closed RNA molecules originate from precursor mRNAs (pre-mRNAs) and are usually from splicing within a protein-coding gene. CircRNAs have several biological functions in normal cells, including acting as sponges to efficiently subtract microRNAs and proteins [36].

CircRNAs can be classified into intronic circRNAs, exonic circRNAs, and exon-intron circRNAs (EIcirRNAs) (Figure 1). Multiple mechanisms have been used to describe the biogenesis of exonic circRNAs, including lariat-driven circularization, RNA binding protein (RBP) mediated circularization, and intron pairing-driven circularization [36]. During splicing, an exon skips, resulting in the back-splicing of RNA folding regions. EIciRNA is formed when the spliced intron lariat, or loop-like structure, remains. The loop-like structure can be mediated by an RBP or be intron-pairing-driven. The final structure is an exonic circRNA when the intron sequence is removed. The intronic circRNAs are formed both upstream and downstream of introns and are mainly accumulated in the nucleus. By contrast, exonic circRNAs without introns are most wielded in the cytoplasm to regulate past-transcriptional gene regulation [37].

## 3. Approaches for Discovering lncRNAs

Most lncRNAs and circRNAs are challenging to discover due to their low expression levels [38]. With the development of innovative technologies, an increasing number of novel linear lncRNAs and circRNAs have been identified by computational analysis of the transcriptomic datasets, high throughput sequencing, and experimental validation. Table 1 lists the techniques readily available for studying linear lncRNA and circRNA expression and function.

### 3.1. ncRNAs Databases

Public databases are one of the most important resources for ncRNAs research. More than 200 public databases are providing comprehensive associations between ncRNAs and their biological functions, including 25 for linear lncRNAs (i.e., TANRIC, CGS, GermlncRNA, LNCat, LncSNP, Lnc2Cancer, lnCeDB, LNCMap, Lnc2Meth, lncATLAS, lncPedia, lncRNAdisease, lncRNome, EVLnRNAs, GreeNC, LNCediting, and lncRNAdb for humans), and 13 for circRNAs (i.e., Circbase, Circnet, CirclncRNAnet, CircRNADb, CIRCpedia v2, DeepBase v2.0, CSCD, Circ2Traits, CircR2Disease, and MiOncoCirc) [36,38,63]. These databases are supported by high throughput sequencing or experimental validation.

### 3.2. High Throughput Identification

Multiple methods provide the systematic expression profiling of lncRNAs (Table 1). Although microarrays by predefined probes are not sensitive enough to identify low expression lncRNAs, microarrays have been used to discover novel ncRNAs at the whole-genome level [64]. The lncRNAs’ sequence may match the previously built microarrays probe sequences and reannotate the initially protein-coding genes to lncRNAs [65]. A microarray, which hybridizes a selected panel of circRNAs explicitly, can be used to detect the annotated circRNAs expression [66]. For example, Shen et al. used a circRNA microarray and identified 262 upregulated and 291 downregulated circRNAs in ovarian endometriomas [67]. RNA-seq is currently the most widespread approach to discover linear lncRNAs and circRNAs. For example, Bi et al. performed RNA-seq experiments on six pairs of ectopic and eutopic endometria samples with endometriosis and identified 952 differentially expressed lncRNAs [68]. The limitations for RNA-seq data include the limited ability to discern linear and circular lncRNAs of similar sequence and the depth of sequencing required for low read count molecules, such as circRNAs [66,69,70]. Many deep sequencing studies on endometriosis have listed lncRNAs, including circRNAs, within the differentially expressed transcripts, but most of the manuscripts focus on protein-coding genes. Wu et al. identified 8660 upregulated and 651 downregulated lncRNAs [71], and Wang et al. detected 146 upregulated and 148 downregulated circRNAs in ovarian endometriosis by high throughput RNA-seq [72].

### 3.3. Experimental Validation

After discovering linear lncRNAs and circRNAs by databases or high throughput methods, experimental approaches can be used to study their expression and function (Table 1). RNA interference (RNAi), antisense oligonucleotides (AOs), and CRISPR systems have successfully been used to knock down lncRNAs [56,57,58,59,60,61,62,73]. However, RNAi and AOs may have non-specific and off-target effects. Further, Goyal et al. found only 38% of lncRNAs were safe to be targeted without deregulating neighboring genes by CRISPR applications [61]. It might be necessary to use multiple strategies to select the best gene silencing approach. Real-time quantitative reverse transcription-polymerase chain reaction (QPCR) is employed to validate the expression of linear lncRNAs and circRNAs [54]. The limitations of QPCR include the need for highly sensitive assays to detect low expression molecules and selecting appropriate endogenous control genes of similar low expression and transcript size. RNA in situ hybridization (ISH) is used to visualize and localize lncRNAs [54]. Holdsworth-Carson et al. performed RNA-seq, RT-PCR, and ISH in endometriosis samples. They found that the long intergenic non-protein coding RNA 339 (LINC00339) is localized in the nucleus of ectopic endometriotic lesions [74]. While ISH allows for the localization of lncRNA, it is generally not quantifiable. Newer technologies, including single-cell RNA-sequencing and spatial transcriptomics, are being used to quantify the expression and localization of protein-coding genes [31,75]. For example, spatial transcriptomics allows both the quantification of expression and localization, but the limited depth of sequencing at this time precludes the identification of low expression molecules [31]. Several groups have used approaches based on protein precipitation to detect key interactions of binding proteins with lncRNAs. Wang et al. used RNA immunoprecipitation to define the LINC00261/miR-132–3p/BCL2L11 regulatory networks [76]. In addition to functional studies, studies correlating expression, localization, and endometriosis phenotype (i.e., pain or infertility, anatomic location of disease, number of adhesions) may significantly impact the field.

## 4. The Importance of Noninvasive Biomarkers in Endometriosis

Laparoscopy remains the gold standard for pathology-confirmed endometriosis diagnosis [77]. While definitive diagnosis and anatomic characterization are critical for appropriate research studies, laparoscopy may not be ideal for all women with suspected endometriosis. Even as a minimally invasive surgical procedure, laparoscopy is associated with high costs, surgical complications, and time away from work and/or family obligations. The accurate diagnosis of visualized endometriosis lesions is surgeon-expertise-dependent [78,79]. Unfortunately, the lesions may not be visible. Biopsies of uterosacral ligaments without visible lesions in women undergoing laparoscopy for pelvic pain revealed 7% with histologically proven endometriosis [80]. Moreover, endometriosis is characterized by a broad panel of different symptoms depending on the localization of the lesions and the characteristics of the patient. Patients often present with intermenstrual bleeding, dysmenorrhea, dyspareunia, dyschezia, dysuria, or chronic pelvic pain. Non-painful endometriosis can be discovered during the surgical evaluation of infertility [81]. Ultrasonography has been proposed as a very good, cost-efficient noninvasive diagnostic tool. Still, this technique is strongly operator-dependent and, even in expert hands, can miss some lesions, particularly superficial lesions [8,82]. Unfortunately, the median time from the onset of symptoms to a diagnosis is seven years, leading to confusion, frustration, and other problems in terms of quality of life. Another aspect that must be considered is that it may be more successfully treated when endometriosis is diagnosed in the earlier stages [10]. Hence, a noninvasive, reliable test is needed to avoid the risks of surgery and shorten the time to diagnosis. Many studies in endometriosis have profiled linear lncRNAs and circRNAs in endometriotic lesions as a means of biomarker discovery, followed by a more focused examination of expression in circulation as a means of minimally invasive diagnosis (Table 2).

The mechanism by which lncRNAs get from endometriotic tissues to circulation may be extracellular vesicles. Extracellular vesicles (EVs) are small membrane-bound vesicles that have emerged as mediators of cell-cell communication by transferring their contents, including lncRNAs [91]. Evidence has highlighted that miRNA, linear lncRNAs, and circRNAs can enter circulation and serve as noninvasive serum biomarkers for diagnosis or prognosis in other diseases, such as lung, colorectal, or prostate cancer [92]. For example, Qiu et al. found that serum extracellular vesicular TC0101441 levels are increased in patients at stage III/IV endometriosis in comparison with stage I/II endometriosis and non-endometriosis control patients. They further showed that this lncRNA played a role in the migration and invasion in cell lines through interaction with metastasis-related proteins, suggesting a possible role in endometriosis pathogenesis [89]. Beyond a functional role for lncRNAs, a potential diagnostic role of EVs lncRNAs has been proposed. Khalaj and collaborators showed a role for nuclear paraspeckle assembly transcript 1 (NEAT1) and H19 imprinted maternally expressed transcript (H19) in the context of a broad interaction with the miR-375, miR–30d-5p, and miR–27a-3p networks [88]. In this study, after the isolation of EVs, an analysis of the contents of the vesicles was performed. The EVs obtained from endometriotic lesions carried a unique miRNA signature compared with the EVs derived from matched patient eutopic endometrium and normal healthy endometrium. Moreover, endometriosis patient plasma-derived EVs carried unique ncRNAs compared with EVs from healthy control eutopic endometrium [88].

ncRNAs promise great results as biomarkers for noninvasive diagnosis purposes because they are resistant to RNase degradation and remain stable in biologic fluids, such as blood (i.e., serum or plasma), saliva, and urine [93]. Recently, circulating linear lncRNAs and circRNAs have been studied in gynecological diseases, gastric cancer, and hepatocellular carcinoma [83,94,95]. Specific to endometriosis, genome-wide profiling determined that a signature-based lncRNA profile, including the lncRNAs NR_038395, NR_038452, ENST00000482343, ENST00000544649, and ENST00000393610, can differentiate patients with and without endometriosis [83]. Notably, this group performed a genome-wide lncRNA analysis with the Glu Grant Transcriptome array in serum samples and eutopic and ectopic endometrium in endometriosis patients and a control group. While the control group had pelvic pain, the control group that had laparoscopy confirmed no evidence of endometriosis. This analysis identified 1682 lncRNAs with dysregulated expression in the sera of patients with endometriosis compared with controls and 1435 lncRNAs in the ectopic endometrium compared with the eutopic endometrium of negative controls. Of these differentially expressed lncRNAs in endometriosis tissues or serum, only 125 were differentially expressed in serum and tissue. After selecting for a similar change in gene expression direction (i.e., down or upregulated), they had a shortlist of 16 lncRNAs. The receiver operating characteristic (ROC) curve analysis used for cross-validation in the study population showed the highest area under the curve (AUC) of a circulating lncRNA was for ENST00000482343. Combining the expression of multiple lncRNAs into a signature-based profile revealed the highest AUC for a panel of NR_038395, NR_038452, ENST00000482343, ENST00000544649, and ENST00000393610. Significantly, the authors correlated the expression of this panel of lncRNAs with clinically relevant laparoscopic features (i.e., pelvic adhesions, ovarian involvement) [83]. A limitation of this study was the lack of external validation of results. The possibility of predicting a challenging surgery with a simple circulating biomarker during the preoperative workup would offer great help to make the right choices in terms of surgeons, surgical team, and surgical equipment. Additionally, women with endometriosis in remote areas without surgeons with expertise in endometriosis surgery could be referred appropriately.

The lncRNA urothelial cancer-associated 1 (UCA1) is another possible diagnostic biomarker for endometriosis. UCA1 was downregulated in the ectopic endometrium of a cohort of 98 endometriosis patients compared to 28 healthy controls. Endometriosis patients were classified with the American Fertility Society (AFS) staging: 19 patients in stage I, 21 patients in stage II, 33 patients in stage III, and 25 patients in stage IV. The relative expression of UCA1 in the serum was lower in women with increased AFS stage. A ROC curve analysis was performed among the study population to evaluate the diagnostic value of serum UCA1. The AUC for Stage I was 0.7509 [95% CI (0.6109 to 0.8910), *p* = 0.003820]; AUC for stage II was 0.9175 [95% CI (0.8308 to 1.004), *p* < 0.0001]; AUC for stage III was 0.9605 [95% CI (0.8982 to 1.023), *p* < 0.0001]; AUC for stage IV was 0.9921 [95% CI (0.9747 to 1.010), *p* < 0.0001]. To evaluate further, circulating UCA1 was examined immediately after surgery and periodically during follow-up. Interestingly, the serum levels of UCA1 were upregulated after treatment and downregulated in cases of relapse. These results suggest that UCA1 is a useful tool for diagnosis and monitoring recurrence [85]. A limitation of this study was the lack of external validation of the results. Similar studies should be performed for other treatment modalities, including medical management.

Up to 50% of women who experience infertility have endometriosis. Studies showed that women with endometriosis have endometrial dysfunction, including progesterone resistance, which may play a role in the timing of endometrial receptivity [96]. Understanding the appropriate timing for embryo transfer may improve pregnancy rates. Studies have examined miRNAs in the eutopic endometrium and peritoneal fluid for infertility evaluation [97,98,99]. Further, an association between endometriosis and some specific ovarian cancer histotypes, particularly endometrioid and clear cell carcinomas, have been shown epidemiologically [100]. Hence, a possible application of peritoneal fluid analysis could help in the early prediction of endometriosis-associated ovarian cancer, as already has been demonstrated for miRNAs [101,102]. Future work in lncRNAs is needed in these areas.

## 5. Therapeutic Opportunities for lncRNAs

As lncRNAs function to regulate gene expression, lncRNAs represent novel therapeutic molecules. Therapeutic noncoding RNAs as targeting molecules, including small interfering RNAs (siRNAs), short hairpin RNAs (shRNAs), miRNA mimics, miRNA sponges, and CRISPR–Cas9-based gene-editing technologies, have been experimentally developed to regulate gene expression and potentially treat disease, but therapeutic targeting using noncoding RNAs is in its infancy [103]. To date, 11 RNA-based therapeutics are approved by the United States Food and Drug Administration (US FDA) and/or the European Medicines Agency (EMA) [104]. While no RNA-based therapeutics are indicated for endometriosis, therapeutic linear lncRNAs and circRNAs may act to inhibit downstream genes and subsequent cellular function and offer significant promise for non-hormonal therapy. Understanding the precise mechanisms of lncRNAs and their antagonists is the first step towards translational applications, as indicated by several preclinical studies highlighted below.

First, the lncRNA H19 imprinted maternally expressed transcript (H19) regulates insulin grown factor receptor (IGF1R) expression by acting as a molecular sponge to let-7 [105]. An in vitro knockdown of H19 with siRNA led to the higher expression of let-7 by real-time quantitative polymerase chain reaction (qPCR) and subsequent inhibition of IGF1R transcript and protein. Functionally, the H19 knockdown resulted in the reduced proliferation of primary endometrial stromal cells isolated from the eutopic endometrium of subjects with endometriosis [105]. Secondly, the molecular sponge mechanism in a preclinical in vitro model can also be found for long intergenic non-protein coding RNA 261 (LINC00261), which binds miR-132-3p and subsequently acts as a regulator of BCL-2-like 11 (BCL2L11) expression. The overexpression of LINC00261 inhibited the proliferation and invasion of the endometriosis cell line CRL-7566 through the BCL2L11 network. Further, the overexpression of LINC00261 revealed a decrease in miR-132-3p expression and increased BCL2L11 expression [76]. The role of BCL2L11 in endometriosis was investigated by siRNA knockdown. BCL2L11 knockdown reduced epithelial-mesenchymal transition (EMT) markers and reduced invasion [76]. While clinically promising, the scientific reproducibility of this effect has not been tested due to the original study being performed in a single cell line, CRL-7566. The CRL-7566 cell line is derived from an ovarian endometrioma. While it was commercially available from American Type Tissue Culture Collection (ATCC), it is no longer available due to its slow growth rate. Further, while the CRL-7566 line was authenticated with short tandem repeat (STR) profiling, it was not well characterized in terms of molecular markers for endometrial epithelium and endometrial stroma [106,107]. Thus, these promising effects need to be replicated.

The in vitro studies on lncRNAs in endometriosis above led to preclinical mouse model studies. First, the lncRNA AFAP1 antisense RNA1 (AFAP1-AS1) mediates the signal transducer and activator of the transcription-transforming growth factor beta-SMAD (STAT3/TGF-β/SMAD) signaling pathway through miR-424-5p to influence endometriosis progression. Huan et al. reported that AFAP1-AS1 knockdown inhibited proliferation and migration and promoted apoptosis in an SV40-transformed, endometriosis eutopic endometrium stromal cell line, hEM15a [108]. Additionally, AFAP1-AS1 regulates EMT. Specifically, AFAP1-AS1 is thought to act in concert with steroid hormones, such as estradiol, to induce the expression of the transcription factor zinc finger E-box binding homeobox 1 (ZEB1). Interestingly, the shRNA knockdown of AFAP1-AS1 reduced the expression of ZEB1 in the spontaneously transformed endometrial cancer cell line Ishikawa. Further, Ishikawa cells with a knockdown of AFAP1-AS2 showed reduced tumor dimensions in a nude mouse model compared to non-targeted Ishikawa cells [109]. These studies highlight the impact of AFAP1-AS1 on proliferation and growth. While promising, the studies in an endometrial cancer cell line highlight the potential lack of clinical applicability to endometriosis. Finally, endometriosis is a disease of significant immunologic features. The use of nude mice, which are immunocompromised, may not be biologically applicable to endometriosis. Improved in vivo and in vitro models are needed to improve the translatability of studies.

Second, Liu and colleagues studied the lncRNA small nucleolar host gene 4 (SNHG4) in a heterologous mouse model of endometriosis. In this model, nude mice were injected subcutaneously with primary endometrial stromal cells (ESCs) isolated from ectopic endometrium and transfected with either NC-si, SNHG4-si1, or SNHG4-si1 combined with anti-miR-148-3p. After silencing SNHG4, the volume of endometriotic lesions was considerably reduced compared to the non-targeting control. Further, the expression of MET proto-oncogene receptor tyrosine kinase (MET) was inhibited, while miR-148a-3p was upregulated. The inhibitor of miR-148a-3p combined with SNHG4 knockdown rescued endometriotic lesions growth and upregulated the MET expression. The authors postulated that SNHG4 might upregulate proto-oncogene expression, in particular MET, via the suppression of miR-148a-3p, to promote the increased growth of endometrial tissue outside the uterine cavity and endometriosis lesions [110]. The impact of oncogenes and the manipulation of oncogenes in therapy for endometriosis deserves future study, particularly as non-hormonal therapies.

Third, studies showed that lncRNA maternally expressed 3 (MEG3-210) has a regulatory mechanism in endometriosis. MEG3-210 was downregulated in the eutopic endometrium of endometriosis patients and the primary cultures of endometrial stromal cells from women with endometriosis. The overexpression of MEG3-210 in the primary cultures of endometrial stromal cells from women with endometriosis revealed reduced invasion and migration. Further, flow cytometry detected a reduction in apoptosis. They examined two molecular pathways, including p38 signaling for its role in the endometriosis inflammatory response and PKA/SERCA2 signaling for its effects on cell motility and apoptosis. Western blotting showed that the protein levels of phosphorylated mitogen-activated protein kinase 14 (better known as p38) and phosphorylated activating transcription factor 2 (ATF2) were significantly increased after the downregulation of MEG3-210. Furthermore, the protein levels of protein kinase cAMP-activated catalytic subunit alpha (PRKACA, better known as PKA) and ATPase sarcoplasmic/endoplasmic reticulum Ca2+ transporting 2 (SERCA2) were decreased after MEG3-210 downregulation [111]. Previously, p38 activity was found to be higher in the eutopic and ectopic endometria in endometriosis patients. Increased p38 MAPK activity in endometriotic cells correlated with the activation of inflammatory cytokines, such as interleukin one beta (IL1b) and tumor necrosis factor-alpha (TNFα) [112]. Finally, they found that p38/MAPK and PKA/SERCA2 signaling pathways act through Galectin1. Galectin-1 is a member of the sub-family of galectins that play a role in intracellular signal processing, molecular modification, cell motility, and malignant biological behavior [111,113]. Recently, MEG3 has regulated transforming growth factor-beta (TGFβ) signaling [114]. Previous work has shown the role of TGFβ signaling in ovarian endometriomas through small RNA signaling [115]. Further studies should examine the connection of lncRNA MEG3 in TGFβ signaling in endometriomas.

Like lncRNA, circular RNAs share the sponge mechanism of action. For example, the circular RNA circ_0007331 targets miR-200c-3p and, consequently, targets hypoxia-inducible factor 1 subunit alpha (HIF1A), a key transcription factor for angiogenesis and hypoxia mechanisms. Through this mechanistic axis, circ_0007331 knockdown, with the cooperation of HIF1A downstream, reduced the proliferation and invasion of primary endometrial cell cultures from women with endometriosis. With the overexpression of miR-200c-3p, proliferation and invasion increased, as did HIF1A. The inhibition of miR-200c-3p, conversely, reduced the proliferation and invasion caused by circ_0007331 knockdown, confirming that the circ_0007331/miR-200c-3p/HIF-1α axis has an important role in cell proliferation and invasion in endometriosis [116]. Using a homologous endometriosis mouse model, treatment with circ_0007331 shRNA, shRNA NC, or anti-miR-200c-3p showed that circ_0007331 knockdown reduced the lesion sizes. Further, treatment with anti-miR-200c-3p did not. Using immunohistochemistry, endometriosis lesions from mice treated with circ_0007331 shRNA were negative for HIF1A, but mice treated with anti-miR-200c-3p treatment maintained HIF1A expression [116]. These results show the importance of the circ_0007331/miR-200c-3p/HIF-1α axis in the endometrium of endometriosis patients.

Finally, the sponge mechanism of action has been proposed for the circular RNA circ_0004712, and miR-148a-3p. Notably, this axis plays an important role in estradiol (E2)-induced EMT processes in the development of endometriosis, potentially through the β-catenin pathway. The E2 treatment of either the endometrial cancer cell line Ishikawa or the human papillomavirus (HPV)-16 E6/E7 transformed endometriosis endocervical cell line End1/E6E7 showed the overexpression of circ_0004712. Further, E2 treatment increased migration in transwell assays and the induction of EMT through the b-catenin pathway. The E2- treatment effect was suppressed with the knockdown of circ_0004712 [117]. Interest in circRNA application in endometriosis is a relatively new area of research. However, the exciting data to date support additional preclinical studies.

While lncRNAs offer opportunities for targeting cellular function, lncRNAs themselves offer options as therapeutic targets. The dysregulation of lncRNA expression has been linked to diseases and complex biological processes [118]. Recently, lncRNA HOX transcript antisense RNA (HOTAIR) has been associated with a genetic susceptibility to endometriosis. Functional single nucleotide polymorphisms, including rs1838169 and rs17720428, were frequently found in endometriosis patients [119]. Moreover, endometriosis pathogenesis may revolve around a functional axis of HOTAIR/homeobox D10 and HOTAIR/homeobox A5. Homeobox proteins (HOXs) are critical in maintaining endometrium homeostasis during embryo implantation and menstrual cycles, highlighting their importance in endometriosis [120]. HOTAIR knockdown reduced cell proliferation and migration and increased HOXD10 and HOXA5 expression in two ovarian clear cell cancer cell lines, ES-2 and TOV-21G [119]. The overexpression of HOTAIR in epithelial ovarian cancer cells increases cancer invasiveness and metastasis. Moreover, the involvement of HOTAIR in cancer progression and response to standard chemotherapy, possibly promoting mesenchymal stem cell formation, has been highlighted [121,122]. Since endometriosis shares features with cancer, these results make HOTAIR a possible target for future endometriosis or ovarian cancer therapies. Secondly, Zhang et al. discovered that another potential target, CCDC144NL antisense RNA1 (CCDC144NL-AS1), was found to be upregulated in ectopic endometriosis and eutopic endometrium from women with endometriosis. The in vitro knockdown of CCDC144NL-AS1 in the SV40-transformed, endometriosis eutopic endometrium stromal cell line hEM15a was associated with decreased migration and invasion. Assuming that alterations in motility and invasion were related to cytoskeleton alteration, the authors found an altered distribution of cytoskeletal F-actin stress fibers compared to lower protein levels of vimentin filaments and matrix metallopeptidase 9 (MMP9) after CCDC144NL-AS1 knockdown [123]. Although there are yet no clinical studies, preclinical studies reveal a potential application of lncRNAs.

In the last few years, a growing interest in diet and nutrition as complementary therapeutic support for endometriosis was established even if randomized clinical trials do not show benefits [124]. Moreover, a connection between nutrition and ncRNA epigenetics has been found with sulforaphane, epigallocatechin gallate (EGCG), genistein, resveratrol, and curcumin in female reproductive tract cancers. A possible therapeutic role of these compounds combined with traditional therapies has been highlighted. As we know, endometriosis shares some pathways with neoplastic disease. Phytochemicals and nutraceuticals have been shown to influence pathways involving the miR-200 family, let-7 family, or miR-34a that can interact with inflammatory and oxidation mechanisms that play an important role in endometriosis [125].

## 6. Challenges to Clinical Application and Future Directions

Linear lncRNAs and circRNAs promise great results as biomarkers for the early detection and disease recurrence of endometriosis. ncRNAs are resistant to RNase degradation and remain stable in biologic fluids, allowing for transport stability to specialized clinical laboratories that may not be local for all women. Studies show promising results but with little consistency among them, especially if considering single lncRNAs as biomarkers. Signature panels of miRNAs, such as the miR-20 or miR-200 families, have been widely investigated but partially have the same problem [126]. A possible solution could be to combine different molecules to obtain a more powerful signature of lncRNAs and miRNAs or other circulating markers (such as CA125) to create a more accurate diagnostic tool.

The main contemporary challenge is the heterogeneity of endometriosis cases and controls. The World Endometriosis Research Foundation (WERF) Endometriosis Phenome and Biobanking Harmonisation Project (EPHect) has provided guidelines [127]. The detailed characterization of women with endometriosis in terms of pain symptoms, lesion location, and molecular profiles is critical to homing in on useful diagnostic tools. While most of the studies interrogate the use of medications, most do not consider nutritional factors, over-the-counter supplements, or drugs. For example, the dietary intake of omega-6 fatty acids, omega-3 fatty acids, vitamin D, and N-acetylcysteine may affect endometriosis development. Further, supplements containing quercetin and L-carnitine may be involved in the progression of endometriosis [128]. Nutraceuticals, nutritional products that are also used as medicines [129], are emerging within the realm of endometriosis therapy [130]. As studies within other gynecologic diseases have shown an effect of nutraceuticals on noncoding RNA expression [125], the role of these natural products, nutrients, and supplements on lncRNAs requires additional study in endometriosis.

Each woman with endometriosis is a unique individual, and small studies are insufficient to evaluate a large number of clinical features. The collaborative, detailed characterization of the phenotype of women with endometriosis is critical. Unfortunately, an optimal non-endometriosis control population is challenging without putting healthy women through laparoscopic surgery for research purposes. While detailed guidelines are helpful for translational studies, additional guidelines are needed to report endometriosis mouse models and in vitro model systems, including multicellular aggregates, spheroids, and organoids. Preclinical studies on lncRNAs and circRNAs show promise for the translation to well-characterized human studies.

## Figures and Tables

**Figure 1 ijms-22-10626-f001:**
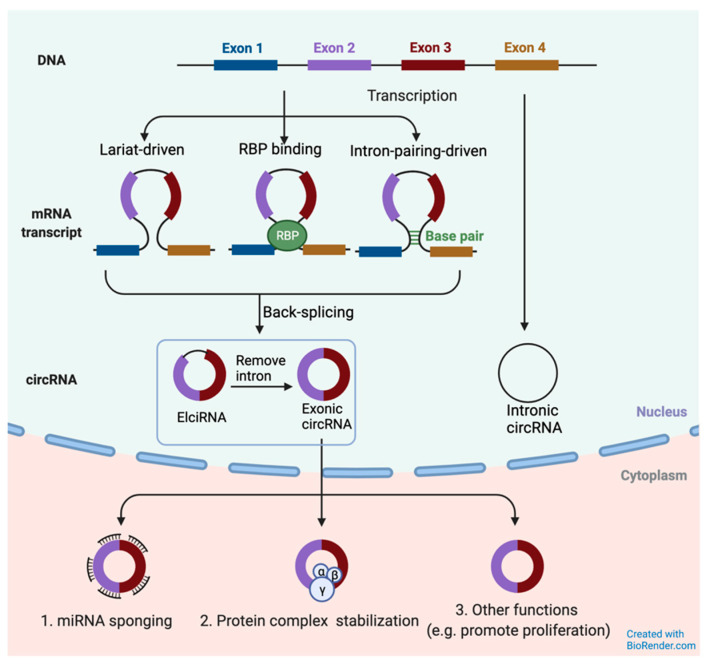
Biogenesis of circRNAs. Three types of circRNAs: exonic-intronic circRNA (EIciRNA), exonic circRNA, and intronic circRNA. EIcirRNAs and intronic circRNAs are abundant in the nucleus. Exonic circRNAs are exported from the nucleus to the cytoplasm and play functional roles in gene transcription and post-transcription. Adapted from [36].

**Table 1 ijms-22-10626-t001:** Key experimental approaches for identifying and validating lncRNA expression and function.

Technique	Description	Ref
Microarray	Identify lncRNAs	[39]
RNA-sequencing	Identify lncRNAs	[40,41]
Tiling arrays	Identify and characterize lncRNAs	[42]
CAGE (Cap analysis of gene expression)	Identify lncRNAs	[43]
ChIRP-seq (chromatin isolation by RNA purification sequencing)	Identify lncRNA-chromatin interactions	[44,45]
CHART-seq (capture hybridization analysis of RNA targets)	Identify lncRNA-chromatin interactions	[46]
3C (chromosome confirmation capture)	Characterize lncRNA-genome binding site	[47]
RAP (RNA antisense purification)	Characterize lncRNA-genome binding site	[48]
RIP-seq (RNA immunoprecipitation)	Identify lncRNA-protein interactions	[49,50]
PAR-CLIP (Photoactivatable-ribonucleoside-enhanced cross-linking and immunoprecipitation)	Identify lncRNA-protein interactions	[51]
RNA pull-downs	Identify lncRNA-protein interactions	[52]
EMSA (Electrophoretic mobility shift assay)	Characterize lncRNA-protein complexes	[53]
RT-qPCR (real-time quantitative polymerase chain reactions)	Cellular localization and expression	[54]
RNA-FISH (RNA-fluorescent in situ hybridization)	Cellular localization	[55]
RNA-ISH (RNA- in situ hybridization)	Cellular localization	[54]
RNAi (RNA interference)	Knockdown lncRNAs	[56,57]
CRISPR-Cas9	Knockdown lncRNAs	[58,59,60,61]
ASO (Antisense oligonucleotides)	Knockdown lncRNAs	[62]

**Table 2 ijms-22-10626-t002:** Summary of the ncRNAs as biomarkers in endometriosis.

Clinical Application	ncRNA	Methods	Number of Patients	Tissue Types	Cycle Phase	Diagnostic Value	Study Type	Ref.
Noninvasive diagnostic biomarkers	NR_038395, NR_038452, ENST00000482343, ENST00000544649, ENST00000393610	Genome-wide transcriptome array	59 endometriosis 51 controls	Case: Endometriosis tissue (eutopic and ectopic endometrium) and blood samples Control: Eutopic endometrium, blood sample	Follicular, 50; luteal, 9 of 59 endometriosis patients Follicular, 44; luteal, 7 of 51 control patients	ENST0000048234 alone: 72.41% sensitivity and 71.74% specificity Panel of NR_038395, NR_038452, ENST00000482343, ENST00000544649 ENST00000393610: 89.66% sensitivity and 73.17% specificity	Case-Control	[83]
Biomarker discovery	86 total differently expressed SNORD3A TCONS_00006582 ABO TCONS_08347373	RNA Sequencing	17 endometriosis 17 controls	Case: Ectopic endometrium Control: eutopic endometrium	Proliferative	Not evaluated	Case-control	[84]
Noninvasive diagnostic biomarkers	UCA1	qRT PCR	98 endometriosis 28 controls	Case: Serum, eutopic-ectopic endometrium Controls: serum	Not evaluated	Stage I: specificity of 80.1% and sensitivity of 76.7% Stage II: specificity of 85.6% and sensitivity of 81.1% Stage III: specificity of 89.1% and sensitivity of 88.1 Stage IV: specificity of 90.5 % and sensitivity of 89.0%	Case-control	[85]
Biomarker discovery	circ_0004712 circ_0002198	CircRNA array	41 endometriosis 22 controls	Case: Ectopic, eutopic endometrium Control: Eutopic endometrium	Proliferative, 28; secretive, 13 of 41 endometriosis patients Proliferative, 16; secretive, 6 of 22 control patients	Not evaluated	Case-control	[86]
Biomarker discovery Therapeutic target discovery	circ_0004712, circ_0002198, circ_0003570, circ_0008951, circ_0017248	CircRNA array	41 endometriosis	Case: Ectopic, eutopic endometrium Control: Eutopic endometrium	Proliferative, 30; secretory, 11 of 41 endometriosis patients	Not evaluated	Discovery	[87]
Noninvasive diagnostic biomarkers	MEG8, SNHG25, LINC00293, LINC00929, RP5-898J17.1, NEAT1, H19	Small RNA sequencing	6 endometriosis Controls none	Case Eutopic ectopic endometrium, plasma, peritoneal fluid (PF)	Not evaluated	Not evaluated	Discovery	[88]
Noninvasive diagnostic biomarkers	TC0101441	FISH qRT-PCR	10 endometriosis 10 control	Case: Ectopic, eutopic endometrium, serum Controls: Eutopic endometrium, serum	Not evaluated	Not evaluated	Discovery	[89]
Noninvasive diagnostic biomarkers	TC0101441	qRT-PCR	29 endometriosis 16 controls	Case: Serum Controls: Serum	Not evaluated	Not evaluated	Discovery	[89]
Noninvasive diagnostic biomarker Recurrence	H19	qRT-PCR	104 endometriosis 50 controls	Case: Ectopic, eutopic endometrium Controls: Eutopic endometrium	Proliferative	sensitivity 90.9% and specificity 61.0%, for predicting recurrence	Case-control	[90]

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
