# Peer review of "Translational Applications of Linear and Circular Long Noncoding RNAs in Endometriosis"

_ijms, 2021, doi:10.3390/ijms221910626_

Round 1
Reviewer 1 Report
The work is well organized and adequately covers a new and interesting topic.
Some points should be better characterized
The role of nutrition in managing endometriosis should be included since nutritional (and nutraceuticals) influence all noncoding RNA expression and action
to be used as biomarkers noncoding RNAs must be secreted in the vesicles, this happens almost always for miRNAs, but not always in the others, this should be better considered; authors might try to suggest a signature-based on present literature.
Similarly, a possible therapy with antagonists would also be more difficult, due to the mechanism still not being well understood and to a greater number of bases than miRNAs.
Author Response
We appreciate the time and effort of the reviewers for their insightful comments listed below. We believe that the additional work, speculation, and clarification requested by the reviewers have made this a significantly better review paper.
Reviewers’ Comments for the Authors are copied verbatim in a non-italic font. Our comments are in blue italic font with marking in the left-hand column of the page. A clean copy of the manuscript is uploaded as the main manuscript. However, a document with a mark-up of changes is included to facilitate the review.
The work is well organized and adequately covers a new and interesting topic.
We appreciate the kind words.
Some points should be better characterized
The role of nutrition in managing endometriosis should be included since nutritional (and nutraceuticals) influence all noncoding RNA expression and action
We appreciate the Reviewer bringing this fascinating field up for our review. While we did not find specific papers on nutraceuticals and lncRNAs in endometriosis, we incorporated some works into our review.
to be used as biomarkers noncoding RNAs must be secreted in the vesicles, this happens almost always for miRNAs, but not always in the others, this should be better considered; authors might try to suggest a signature-based on present literature.
We have added specific wording regarding a signature-based lncRNA profile from the literature.
Similarly, a possible therapy with antagonists would also be more difficult, due to the mechanism still not being well understood and to a greater number of bases than miRNAs.
Therapy comes with challenges. Within the review, we described two different means to use lncRNAs as therapeutic molecules. One way is to use lncRNAs to target downstream effector signaling molecules and pathways. The second way is to use molecular targets to lncRNAs where the function is known. We agree that additional research is needed, and we have clarified this wording.

Reviewer 2 Report
In this manuscript, the authors did a comprehensive literature review to summarize the role of long noncoding RNAs (lncRNAs) and Circular RNAs (CircRNA) in endometriosis, to address the situation that a definitively diagnosis for endometriosis requires invasive surgical procedures, with the risk of complications, additional surgeries in the future, and a high rate of recurrence, however no-invasive diagnostic strategies for endometriosis are not well-studied.
The manuscript is well-written. A comprehensive literature review was carried out and the every aspects related to the subjects were logically described in order.
The introduction section briefly described the current knowledge on Endometriosis as well as diagnostic and therapeutic strategies, and basic information of ncRNAs. The last section (6) provides a clear narrative of the potential importance as well as challenges to clinical application of lncRNAs and CircRNA in diagnosis and therapy of endometriosis, and proposed future directions. I only have few minor suggestions.
(1) CircRNAs are a special subtype of the lncRNAs that are covalently closed, single-stranded circular transcripts without 5’ caps or 3’ poly(A) tails. I'm wondering if the title of this manuscript could be changed to 'The Role of Long Noncoding RNAs in Endometriosis'? Reader may have an impression that lncRNAs and CircRNAs are essentially independent. This may also be applied in other parts of manuscript where applicable.
(2) In the Introduction section, I would suggest a little bit more information on non-invasive diagnosis other than lncRNAs and CircRNAs, although there is a independent section about this. Generally, if you would like to propose a potential novel strategy, you may want to briefly list out currently available methods and summarize limitations.
(3) In 3. Approaches for discovering lncRNAs and circRNAs, the authors need to summarize which approach(s) is/are more generally used and the pros/cons of them.
(4) Across the manuscript, when AUC was presented, it should be explicit whether they were from cross-validation or external validation.
Author Response
We appreciate the time and effort of the reviewers for their insightful comments listed below. We believe that the additional work, speculation, and clarification requested by the reviewers have made this a significantly better review paper.
Reviewers’ Comments for the Authors are copied verbatim in a non-italic font. Our comments are in blue italic font with marking in the left-hand column of the page. A clean copy of the manuscript is uploaded as the main manuscript. However, a document with a mark-up of changes is included to facilitate the review.
In this manuscript, the authors did a comprehensive literature review to summarize the role of long noncoding RNAs (lncRNAs) and Circular RNAs (CircRNA) in endometriosis, to address the situation that a definitively diagnosis for endometriosis requires invasive surgical procedures, with the risk of complications, additional surgeries in the future, and a high rate of recurrence, however no-invasive diagnostic strategies for endometriosis are not well-studied.
The manuscript is well-written. A comprehensive literature review was carried out and the every aspects related to the subjects were logically described in order.
The introduction section briefly described the current knowledge on Endometriosis as well as diagnostic and therapeutic strategies, and basic information of ncRNAs. The last section (6) provides a clear narrative of the potential importance as well as challenges to clinical application of lncRNAs and CircRNA in diagnosis and therapy of endometriosis, and proposed future directions. I only have few minor suggestions.
We appreciate the kind words.
(1) CircRNAs are a special subtype of the lncRNAs that are covalently closed, single-stranded circular transcripts without 5’ caps or 3’ poly(A) tails. I’m wondering if the title of this manuscript could be changed to ‘The Role of Long Noncoding RNAs in Endometriosis’? Reader may have an impression that lncRNAs and CircRNAs are essentially independent. This may also be applied in other parts of manuscript where applicable.
We thank the Reviewer for noting this subtle nuance in nomenclature. We have clarified and defined lncRNAs and circRNAs as a subset of lncRNA throughout. We have also changed the title to reflect this.
(2) In the Introduction section, I would suggest a little bit more information on non-invasive diagnosis other than lncRNAs and CircRNAs, although there is a independent section about this. Generally, if you would like to propose a potential novel strategy, you may want to briefly list out currently available methods and summarize limitations.
We have added brief details regarding the available diagnostic methods and their limitations.
(3) In 3. Approaches for discovering lncRNAs and circRNAs, the authors need to summarize which approach(s) is/are more generally used and the pros/cons of them.
We have added details regarding the limitations and benefits of the more common approaches used for genomic-wide discovery. We have added limitations for validation studies on expression. We have added details of functional validation studies.
(4) Across the manuscript, when AUC was presented, it should be explicit whether they were from cross-validation or external validation.
We agree on this critical point. We have clarified the cross-validation design and noted the limitations of lack of external validation as appropriate.
